# Chromatographic-Based Platforms as New Avenues for Scientific Progress and Sustainability

**DOI:** 10.3390/molecules27165267

**Published:** 2022-08-18

**Authors:** José S. Câmara, Cátia Martins, Jorge A. M. Pereira, Rosa Perestrelo, Sílvia M. Rocha

**Affiliations:** 1CQM-Centro de Química da Madeira, Universidade da Madeira, Campus da Penteada, 9020-105 Funchal, Portugal; 2Departamento de Química, Faculdade de Ciências Exatas e Engenharia, Universidade da Madeira, Campus da Penteada, 9020-105 Funchal, Portugal; 3Departamento de Química & LAQV-REQUIMTE, Universidade de Aveiro, Campus Universitário Santiago, 3810-193 Aveiro, Portugal

**Keywords:** gas chromatography, liquid chromatography, multidimensionality, detectors, major achievements, science advances, societal challenges

## Abstract

Chromatography was born approximately one century ago and has undergone outstanding technological improvements in innovation, research, and development since then that has made it fundamental to advances in knowledge at different levels, with a relevant impact on the well-being and health of individuals. Chromatography boosted a comprehensive and deeper understanding of the complexity and diversity of human–environment interactions and systems, how these interactions affect our life, and the several societal challenges we are currently facing, namely those related to the sustainability of our planet and the future generations. From the life sciences, which allowed us to identify endogenous metabolites relevant to disease mechanisms, to the OMICS field, nanotechnology, clinical and forensic analysis, drug discovery, environment, and “foodprint”, among others, the wide range of applications of today’s chromatographic techniques is impressive. This is fueled by a great variability of powerful chromatographic instruments currently available, with very high sensitivity, resolution, and identification capacity, that provide a strong basis for an analytical platform able to support the challenging demands of the postgenomic and post COVID-19 eras. Within this context, this review aims to address the great utility of chromatography in helping to cope with several societal-based challenges, such as the characterization of disease and/or physiological status, and the response to current agri-food industry challenges of food safety and sustainability, or the monitoring of environmental contamination. These are increasingly important challenges considering the climate changes, the tons of food waste produced every day, and the exponential growth of the human population. In this context, the principles governing the separation mechanisms in chromatography as well the different types and chromatographic techniques will be described. In addition, the major achievements and the most important technological advances will be also highlighted. Finally, a set of studies was selected in order to evince the importance of different chromatographic analyses to understand processes or create fundamental information in the response to current societal challenges.

## 1. Introduction

Chromatography, which means *to show with colors*, is certainly among the most important analytical procedures supporting science and human development in the last century. Chromatography is present in the most diverse fields of human activity, spanning from science to the pharmaceutical, chemical, and food industries and the areas of health and the environment. According to the IUPAC definition [1], chromatography is a physical method of separation in which the components to be separated are distributed between two phases, one of which is stationary (stationary phase) while the other (the mobile phase) moves in a definite direction. In scientific terms, chromatography is not one, but several related techniques able to separate mixtures and it is so powerful that it can be used to separate proteins differing in a single amino acid, compounds differing only in the spatial orientation of a functional group or a volatile compound in a mixture. Eventually, one of the forms to assess the importance of chromatography is to observe the number of articles published in the literature involving this technique. Accordingly, over 736,000 articles have been published since 1941, more than half of these after 2000, with an average of over 20,000 per year in the last three years. These numbers certify the importance of chromatography in the different fields and research, but also spanning the most diverse human activities in everyday life, being used as a tool to control the quality of everything we interact with, including what we eat or drink, where we do it, what we wear, and even the quality and safety of the air we breathe, both inside and outside of the places we live and work and in our cities, many of them already deeply affected by poor air quality, seriously affecting susceptible persons.

This review intends to give the reader an overview of the concepts related to the different types of chromatography, and the developments from its invention approximately 100 years ago up to the present time, to respond to the societal challenges and legislation requirements. Several examples were selected to highlight the immense importance of chromatographic analysis, directly and indirectly, in our lives and the understanding of our surroundings. In fact, instrumental developments have been catalyzed by the growing challenges of a society in constant change, in which higher specificity, sensitivity, and resolution are continually sought.

### 1.1. History and Evolution

German chemist Friedlieb Ferdinand Runge, the scientist that discovered caffeine and quinine, was also the first to use chromatography principles in the early 1800s (Figure 1). Runge was interested in the production of dyes and bleaches from coal tars, and he used filter paper to obtain color separations from those mixtures [2,3]. The color patterns that Runge obtained (self-grown pictures) are considered an incipient form of paper chromatography, despite Runge not describing any theoretical considerations to support his findings. In 1861, Groppelsroede used the capillary properties of water to separate colored pigments. To perform this capillary analysis, he dipped one end of the paper strip into an aqueous solution [3,4]. Groppelsroede, similarly to Runge, did not report a satisfactory explanation for such separation resembling paper chromatography. In the early 1900s, however, Mikhail Tsvet used a hydrocarbon solvent and a carbohydrate powder as a stationary phase to separate chlorophylls and plant pigments from green leaves. The colored bands Tsvet obtained certainly led him to name the technique he invented chromatography, in that case, adsorption chromatography. Curiously, his surname, Tsvet, means color in Russian. Tsvet has made hundreds of experiments and he was the first scientist to explain the fundamentals of chromatography [5] (Figure 1). Most of the principles Tsvet described are still applicable nowadays in modern chromatography. For that reason, he is known as the father of chromatography. Unfortunately, the seminal work of Tsvet was only published in Russian, and that prevented its fast dissemination. For that reason, the next hallmark in chromatography development did not occur until almost three decades later. In 1931, Lederer and collaborators achieved the separation of lutein and zeaxanthin in carbon disulphide and the xanthophylls from egg yolk on a column of calcium carbonate powder 7 cm in diameter [6]. Several years later, flow-through chromatography became an important laboratory separation technique. The work of Khun, Karrier, and Ruzicka, awarded with the Nobel Prize in 1937, 1938, and 1939, respectively, for their work on chromatography, was another major hallmark of chromatography evolution. The next important milestone, eventually one of the most important, was made by biochemists Archer Martin and Richard Laurence Millington Synge, awarded the Nobel Prize in Chemistry in 1952 for the invention of partition chromatography. The original process developed by Martin and Synge to isolate acylated amino acids from protein hydrolysates by extraction of the aqueous phase with a chloroform organic phase was very tedious, but it boosted future and unprecedented improvements in chromatographic resolution. Soon, Martin’s and Synge’s proposal was improved with a chromatographic column containing silica gel particles and later cellulose [7] (Figure 1). Since then, chromatography developed rapidly as well as in several directions beyond liquid–liquid and gas–liquid partition chromatography. During the 50s, for instance, the paper was substituted by thin layers of silica gel, originating the thin-layer chromatography (TLC), and the initial simple and well-defined mixtures containing small molecules gave rise to more complex biological systems. At another level, protein chromatography underwent a significant evolution using technologies such as reverse-phase, hydrophobic interaction, and affinity chromatography.

In its initial development, column liquid chromatography was hampered by the very slow separation process caused by mobile phases crossing 100 µm sorbent particles in the column only by gravity. To achieve faster separations, Csaba Horváth and Seymour Lipsky of Yale University introduced the use of pumps. In 1966, they published a paper describing an ion-exchange separation of organic compounds [8], and one year later they introduced fast liquid chromatography (LC) using a pump operating at higher pressures than in their previous work [9]. Horváth is generally credited with building the first high-pressure liquid chromatography instrument [10]. Furthermore, effectively during the decade 1970, the introduction of pumps allowed a major increase in the mobile phases flow and the definition of high-pressure liquid chromatography [5]. Meanwhile, durable adsorbents with much lower particle sizes between 5 and 10 μm, high-pressure pumps up to 400 atm, and flow-through detection systems were developed, and high-pressure was substituted by high-performance liquid chromatography (HPLC), which is how the HPLC acronymous is still currently used [5]. Soon, HPLC surpasses gas chromatography (GC) in terms of applications and importance. This period was known as the renaissance of LC [5]. In the late 1970s, GC took advantage of the development of fused silica capillaries, which dramatically improved its performance and efficiency.

However, GC was limited to the analysis of volatile compounds and many important analytes could not be derivatized and made volatile. Therefore, by the 1980s, HPLC was already used in most laboratories worldwide and progressive improvements in sensitivity, speed, accessibility, and resolution continue to be made. The capacity to separate very similar compounds, including chiral compounds, should also be mentioned as an important achievement of chromatography as an analytical tool. Chiral chromatography was boosted by the work of Davankov with chiral ligand exchange chromatography, a method for separating optical racemate isomers [11].

The introduction of new detectors, first using UV/IR spectrophotometry, and then mass spectrometry (MS), was another hallmark in chromatography evolution. The use of MS detectors allowed a major improvement in the detection limits and identification of compounds in many fields of research, particularly in protein analysis.

Ion chromatography (IC) also met considerable improvements, the most notable the use of the suppressor. This strategy removes the eluent, therefore, limiting the background and reducing noise, and favors the conversion of the analytes into more conductive forms, resulting in a major gain in sensitivity of the IC.

Additionally, very relevant are the developments of particle technology and column chemistries, from the initial big and fully porous particles to the superficially porous materials, currently under 2 µm in diameter. This was essential to introduce new column features, lower dispersibility, and enhanced binding capacity to respond to the high throughput, speed, and resolution requirements of the biopharmaceutical industry. The successive developments in chromatography requested faster and more powerful processing units and connections and this would be not possible with the development of faster computers and internet connections to process and share the growing amount of data. The first computers and software to work with chromatography were only introduced in the late 1980s and they were a remarkable evolution. Modern electronic devices and chromatographic configurations can produce, process, and transfer data at collection rates of up to 200 Hz. This means thousands of samples being processed each day, something that was unimaginable only a few years ago. Figure 1 condenses some of the most important features and hallmarks in chromatography since its invention.

Currently, HPLC is almost omnipresent in any field of research and industry and continues to evolve toward miniaturized, faster, and more efficient separations, using more comprehensive and sensitive analytic sensors, and user-friendly analytical platforms to push the limits of high-resolution and high-throughput even further. Capillary GC or simply GC is also still frequently used, but only for volatile and semivolatile analytes, while the other forms of chromatography, despite some advances, such as the high-performance TLC (HPTLC), are used to a lesser extent.

Chromatography has been essential in establishing many important therapies for patient care and, certainly, this will be a field where it will continue to evolve. New treatments involving single cell and gene therapies are at the forefront of healthcare and chromatography will be fundamental to bringing new drugs and treatments to the market faster than any other analytical procedure can deliver [10].

### 1.2. Principle of Chromatography and Types of Chromatography

As an analytical technique, chromatography has contributed to facing and solving various societal challenges, being used to separate, isolate, and purify the components of a mixture for qualitative and quantitative purposes. Chromatography is a biophysical-chemical approach of separation where the target analytes in a mixture are smeared onto a solid in the form of a porous bed, bulk liquid layers, or films (stationary phase) and separated with the help of a mobile phase (e.g., liquid, gaseous) that percolates through the stationary phase. During the migration of the fluid through the stationary phase, different transport phenomena, which include diffusion and flow anisotropy, will occur. The relative interaction of a solute with these two phases is expressed by the partition (K) or distribution (D) coefficient, which corresponds to the ratio between the concentration of solute in the stationary phase and the concentration of solute in the mobile phase. These differences result in the retention of some components in the stationary phase that will move slowly in the chromatography system [12].

Chromatography can be classified according to the mobile phase, the shape of the chromatographic bed, or the stationary phase, with the mobile phase being the most important criterion since it defines the class of samples that can be investigated with each one of the chromatographic versions (Figure 2).

Consequently, a division can be made among LC, supercritical fluid (SFC), and GC. Chromatography can also be defined according to four different sorption mechanisms, namely surface adsorption, partition, ion exchange, and size exclusion (Figure 3). In the adsorption process, the stationary phase is a polar adsorbent, normally silica (slightly acidic) but also alumina (slightly basic), charcoal (nonpolar), or several other materials, while the mobile phase is nonpolar (usually a solvent with polarity within the range from hexane to esters). The mobile phase competes with the sample analytes for adsorption at the active sites of the stationary phase. Due to weak interactions with the stationary phase, nonpolar analytes are eluted first, followed by analytes of increasing polarity. It is the method of choice for isomer separation due to the steric properties of the sample compounds. The intermolecular forces primarily responsible for chromatographic adsorption include van der Waals forces, electrostatic forces, hydrogen bonds, and hydrophobic interactions [13].

The partition chromatography mechanism is based on a thin film formed on the surface of a solid support by a liquid stationary phase. Solute equilibrates between the mobile phase and the stationary liquid. In ion exchange chromatography, the ion exchanger groups can be bonded to silica or polystyrene. It is based on ionic equilibria between solute, buffer, and stationary phase ions and counter-ions.

In size exclusion chromatography, the sample analytes will be separated according to their size. They will not be retained by the column packing but will enter the pores where the mobile phase is stagnant. Large molecules can use a smaller fraction of the pores’ volume than small molecules and will be eluted earlier. Analytes that are bigger than the pores are excluded and will appear as the first fraction at the column end [12].

Nevertheless, a suitable sample preparation procedure prior to chromatography analysis is crucial, since direct analysis can result in low sensitivity, accuracy, and reproducibility due to the existence of interferents in the sample matrix. In this sense, to achieve high-quality analytical data with high accuracy, reproducibility, selectivity, and low sensitivity limits, the sample preparation procedure should include steps such as fractionation, isolation, and enrichment of the target analyte. Such an endeavor has been achieved through the improvement of the properties of existent materials, as well as the discovery of new ones, such as ionic liquids, graphene-derived materials, molecularly imprinted polymers, magnetic nanoparticles, carbonaceous nanomaterials, among others.

#### 1.2.1. Gas Chromatography

GC is a popular analytical approach used to separate and analyze samples that can be vaporized (vapor pressure at temperatures below 350–400 °C) without thermal decomposition. In GC, the components of a sample are dissolved in a solvent and vaporized to separate the analytes by distributing the sample between stationary and mobile phases. The mobile phase is a chemically inert gas (e.g., helium, hydrogen, nitrogen) that transports the target analytes through the heated column. A constant flow rate of carrier gas is crucial since it significantly influences the separation efficiency and the quantification of results [12].

Packed and capillary (also known as open tubular) columns with different dimensions are the most used columns for GC. Packed columns comprise finely divided, inert, solid supports (e.g., diatomaceous earth, fluorocarbons, graphitized carbon black, and glass beads) coated with the liquid stationary phase. Packed columns present advantages compared to capillary columns since they have between 10 and 1000 times better sample capacity, which requires a large amount of sample. Nevertheless, packed columns have from 25–50% fewer theoretical plates per meter (m) than capillary columns. A capillary column is a glass or fused-silica tube of very small internal diameter (usually between 0.20 and 0.53 mm). The inner surface of a capillary column is coated with a thin layer of stationary phase, so it is still possible for the solute molecules to contact the inner walls of the tubing. The high separation efficiency is one of the advantages of a capillary column, allowing peak resolution. In addition, the separation efficiency can be increased through the application of a temperature gradient instead of an isothermal separation mode. After the elution of the column, the target analytes can be detected by several detectors, such as mass spectrometry (MS), flame ionization detector (FID), electron capture detector (ECD), nitrogen and phosphorous detector (NPD), among others [14,15]. FID detectors are usually employed in portable GCs. These configurations are used to separate and analyze target analytes which can be vaporized without decomposition. Overall, FID detectors present several advantages such as reliability, versality, ease of operation, and simplicity. In addition, FID shows small or no signal for common carrier gas (e.g., He, Ar, N_2_) or common contaminants (e.g., O_2_, H_2_O). The main drawback of these detectors is its destructive nature and inability to provide structural information of target analytes. The MS detector is another detector often coupled to GC, allowing to elucidate the molecular mass and molecular structure of identified and/or quantified compounds. MS can be used without coupling to GC, but the interpretation is more difficult, mainly when complex mixtures are analyzed. To overcome this problem, MS can be used in tandem with other MS detectors.

#### 1.2.2. Liquid Chromatography

The mobile phase of LC is a liquid (organic solvent, water, and/or a solution of two or more organic solvents and water) and there are several configurations regarding the instrumental as well as the separation steps, such as paper, TLC, and HPLC.

Paper chromatography is the simplest and most cost-effective LC technique, in which the chromatographic bed comprises a paper (e.g., cellulose) shoaled in a liquid solvent that acts as a mobile phase. The samples are applied a few centimeters from the end of the paper and impregnated with some suitable solvent. The solvents penetrate the paper through capillary action (depending on whether ascending or descending development is applied), and, in passing over the sample spot, transport along with it the components of the sample. Those components move with the flowing solvent at velocities that are dependent on their solubilities in the stationary and flowing solvents. In the end, the paper is removed from the developing chamber and the separated zones are detected by applying a suitable method [12].

The TLC is more versatile than the paper chromatography used to separate non-volatile compounds, since several different stationary phases are available, such as cellulose, silica, and alumina. The separation is based on the affinity between the target analytes and the adsorbent that will appear as individual spots on the paper after the chromatographic separation. TLC is an analytical approach broadly used since it is simple, cost-effective, fast, and has good reproducibility and high sensitivity [16]. In addition, high-performance TLC, designed as HPTLC, that uses 5 and 10 μm stationary phase particles, allow for higher separations.

HPLC is an up-to-date application of LC. HPLC pumps the target analyte dissolved in a solvent (mobile phase) at high pressure through a column with an immobilized chromatographic packing material (stationary phase). The retention time of the target analyte depends on the sample and solvent properties, as well as the stationary phase. Thus, the analytes that have the highest interaction with the stationary phase exit the column at the end and, consequently, they display higher retention times.

Diverse solvent combinations can be used as mobile phases and can also contain water and/or organic components. The HPLC column is frequently a stainless-steel tube ranging from 50 to 250 mm in length and from 1–4.6 mm in diameter, packed with chemically modified silica particles (<1–5 μm in diameter). The microparticle packing (low diameter compared to conventional HPLC columns) contributes to the improved resolution of the mixture, requires low consumption of the mobile phases and samples, and can easily be used in tandem with MS due to the lower flow rates involved. There are several stationary phases available for HPLC separations, with silica particles modified with C18 groups the most used. In addition, the diversity of column diameters, length, particle size, and solvent selection contributed to the huge number of choices commercially available to the researcher [17].

HPLC analyses can be carried out in a diversity of modes: (i) reverse phase chromatography (the most important), in which the stationary phase is nonpolar, and in most cases hydrophobic (C18-modified silica); (ii) normal phase chromatography, where the stationary phase is hydrophilic (silica); (iii) hydrophilic inter-action chromatography; (iv) ion chromatography, where the stationary phase is an ion exchange material for cationic or anionic analytes. The mechanisms responsible for distribution between phases include surface absorption, ion exchange, relative solubilities, and steric effects. After the elution from the column, the target analytes can be detected by several detectors, such as MS, photodiode array (PDA), fluorescence (FLD), refractive-index detector, among others. The main disadvantages of PDA and FLD are the restricted number of target analytes with absorbance or fluorescence properties, and the limited sensitivity which the low injection volumes applied allow to obtain. To overcome this problem, LC is often used tandem with MS. This detector has the advantage of unequivocal mass identification and provides structural information of the target analytes. This coupling was made possible due to the development of atmospheric pressure ionization (API) sources, such as electrospray ionization (ESI) and atmospheric pressure chemical ionization (APCI). HPLC has the advantages of high sensitivity, accuracy, separation efficiency, and wide application range, particularly for the detection of high boiling point and non-volatile compounds [12].

### 1.3. Major Achievements

One of the most crucial changes in chromatography is the reduction of time analysis without losing the resolution. In this sense, fast-GC appears to have improved their predecessors, namely concerning rapid oven heating and cooling, extended inlet pressures, support for hydrogen carrier gas, and faster detector response time. This was possible because several mechanisms were adopted in fast-GC, such as adjusting column dimensions, higher flow of carrier gas, lowering the pressure at the chromatographic outlet and increasing the heating temperature in the programmed temperature [18]. Perez-Palacios et al. [19] established the optimum combination of parameters (methanol + chlorotrimethylsilane, lyophilized samples, and oven heating) to achieve the quantification of the highest possible amount of fatty acids in meats, reducing the time of a GC run from 60 to 10 min. Paolini et al. [20] developed a fast GC–FID method for the analysis and quantification of 16 volatile compounds belonging to different chemical classes in alcoholic beverages, using a chromatographic run of only 8 min. Fialkov et al. [21] achieved reasonably good separations with full analysis cycle times of less than 1 min by combining, for the first time, low-pressure (LP) GC–MS with low-thermal mass (LTM) resistive-heating for rapid temperature ramping and cooling of the capillary column. This column configuration in LTM-LPGC-MS trades a 64-fold gain in speed of analysis versus standard GC–MS and a 4-fold loss in chromatographic peak capacity, thereby converting analysis time from minutes into seconds in common applications.

Regarding LC, the reduction in the internal diameter of the column as well as of the particle size of the stationary phase contributed to improving the separation of several target analytes. Ultra-high performance liquid chromatography (UHPLC) appeared as a greener approach to LC, since it requires a lower solvent consumption and offers greater chromatographic resolution and higher sensitivity while requiring a shorter analysis time [22]. Zhao et al. [23] compare the efficiency of HPLC–MS/MS and UHPLC–MS/MS analytical platforms in the simultaneous determination of eight coccidiostats in beef. The data obtained showed that UHPLC–MS/MS was more efficient, faster, and consumed less mobile phase than HPLC–MS/MS. In addition, despite the eight target analytes could be completely separated from impurities using the two methods, the sensitivity of the UHPLC–MS/MS was higher than that of HPLC–MS/MS. The Brazilian green propolis phenolic profile was established using UHPLC-ESI-QToFMS and HPLC–MS. The data obtained showed, for the first time, three different isomers of isochlorogenic acid identified using UHPLC-ESI-QToFMS, indicating the precision and accuracy of this analytical approach [24]. More recently, miniaturized LC versions, such as microflow LC (internal diameter 0.5–1 mm), capillary liquid chromatography (100–500 μm), and nano liquid chromatography (NanoLC, <100 μm), improved sensitivities in comparison to LC [18]. Ponce-Rodríguez et al. [25] assessed the performance and operability of a portable NanoLC in the determination of several methylxanthines in water samples. In addition, a comparative study was also performed between in-tube solid-phase microextraction (IT-SPME) coupled with capillary liquid chromatography (CapLC) or NanoLC. The data obtained showed that IT-SPME/portable NanoLC-based methods were much better in terms of chromatographic resolution and organic solvent consumption per sample, using only 200 μL versus 10 mL for IT-SPME-CapLC. Piendl et al. Of note, Piendl et al. [26] introduced the online hyphenation of chip-based high-performance liquid chromatography (chipHPLC) with ion mobility spectrometry (IMS) via fully integrated electrospray emitters. The method showed promising results, highlighting the potential of IMS as a detection technique for chip configurations, adding the advantages of simplicity, portability, economy, and robustness. Lam et al. [27] proposed a hand-held, battery-powered, portable LC system with a high sensitivity UV-LED-based z-cell detection (HSDC) and compared it with on-column detection strategies used in previous studies. The HSDC flow-cell-based detector showed excellent sensitivity, with low stray light levels, and negligible heat effects using low input currents (<20 mA).

Another challenging trend to improve the separation power is multidimensional chromatography, with GC × GC and LC × LC the most popular. Nevertheless, one of the main drawbacks of this improvement compared to other chromatography approaches (e.g., fast-GC) is the time of analysis required [13]. Multidimensional chromatography is an analytical approach which can deliver heightened separation performance for complex and difficult substances. This is obtained by passing the sample through two different separation stages, applying multiple columns, each with a different stationary phase. Whiting et al. [28] described the development and evaluation of micro-comprehensive two-dimensional gas chromatography (GC × GC) constituted by microfabricated columns and a nanoelectromechanical system resonator as a detector. The resulting system allows eco-friendly and ultrafast chromatographic separation. Acevedo et al. [29] described two-dimensional separation (LC × LC) pioneering applying sequential injection chromatography (SIC) to enhance the performance of the analysis of aromatic biogenic amines. The proposed analytical approach led to a cost-effective, greener, and faster chromatographic separation without involving additional mobile phases and energy consumption. However, in multi-dimensional LC, the robustness of the method is not very good, since the eluting portion of the first column was not fully compatible with the second column, resulting in peak dispersion and retention time shifts.

Finally, it is interesting to note that the basic principles that determines the separation of analytes in the simplest chromatography modes, such as paper and TLC, are the same principles that determine the separation in high-resolution and high-performance chromatography approaches, such as GC × GC-ToFMS or the UHPLC-QToFMS. However, the technological evolution from the former to the recent techniques is immense, namely in terms of the nature and particle size (submicron) of the stationary phases, enabling highly efficient separations, with high-resolution power and short analysis time. Such configurations also involve efficient detection systems with extremely low sensitivities that allow the detection of trace levels as low as femtomole. In addition, the evolution of chromatography data systems (CDSs) in the last decades also potentiated even further the contribution of chromatography to the progress of several fields of science. In fact, the significant evolution of CDSs in the last decades is remarkable. In the 1970s, we were in the “stone age”, where the chromatographic signals were recorded on chart paper followed by peak quantification using manual peak measurement, by “cutting and weighing” the actual peaks. In contrast, in the current “technologic age”, the automation of peak measurement and instrument control, and CDSs data upload and storage in readily shareable online databases is a reality.

## 2. Contributions of Chromatography to the Science Progress

Science has had a significant impact on the understanding of various phenomena associated with the evolution of humanity supported by a remarkable and unique technological evolution, namely from the fourth industrial revolution (4IR), which boosted the use of the internet of things (IoT), modern smart technology, and large-scale machine-to-machine communication (M2M). Chromatography also played a pivotal role in this accelerated development, regularly providing us with new technical and technological advances that boosted the advancement of various areas of science in the post-genomics era, from clinical to agricultural and food sciences, including pharmaceuticals, environment, and health sciences, namely in the diagnosis of various pathologies, in the assessment of therapeutic efficacy and progression of the individual’s state, and, more recently, in the metabolomics field (Figure 4).

It has indeed had a major impact on the evolution of science, and it will also be important to create resilient innovation pathways. It allows us to have a healthier, cleaner, and wealthier world. Its robustness and very high-resolution power associated, namely, with the most recent instruments, help us to better understand and look for the building blocks of life, the Universe, and the possibility of life on other planets, which is the reason why chromatographic systems have been sent on different space missions such as Apollo (moon), Cassini–Huygens (Saturn), Rosseta (comets), and Viking and Curiosity (Mars) [30,31,32].

In fact, chromatography has a strong impact on our everyday life and its contribution to the progress of science is unquestionable, namely, when hyphenated with MS and other high-resolution analytical techniques. The technologic chromatographic advances in terms of columns and detection systems, made it possible to detect and quantify trace amounts of compounds affecting the very basics of life processes.

Currently, the usage of chromatographic techniques is mainly driven by the hospitals and R&D laboratories, the agricultural and food industry, pharmaceutical and biotechnology, the environmental industry, and the petrochemical industry (Figure 4). Much of the advances and progress in health sciences and medicine are directly related to technological advances in chromatography and related separation techniques. The availability of equipment with high-resolution power, sensitivity, and identification made possible a more comprehensive and in-depth knowledge of the metabolic alterations of our organism that allow us to differentiate pathological from healthy states [33,34,35]. The identification of cancer cells, the finding of the most effective antibodies for neutralizing the deadly Ebola virus, and determining which antibodies fight various diseases and viruses [36,37], in addition to the alterations of metabolic pathways determined for different pathologic status, exemplify some applications of the importance of chromatography in the advancement of medical science supported in the early diagnosis of certain pathologies [38], and in the evolution of therapeutic efficacy [39], with personal, social, and economic impact.

In the food science and food industry, chromatographic techniques are used for the establishment of the “foodprint” through the identification and characterization of different food components, including the volatile compounds, some of them associated with the aromatic and organoleptic characteristics of the food [40], vitamins [41], proteins [42], amino acids [43], mono-(MUFAs) and polyunsaturated fatty acids (PUFAs) [44], carbohydrates, polysaccharides [45], and contaminants [46] together with their metabolism, toxicology, and food fate. Additionally, chromatographic techniques have proved to be powerful tools in defining the authenticity and typicality of foods [47], helping to protect them from potential frauds and/or adulterations. In this context, we can refer to an example, the horsemeat scandal in 2013, which positioned chromatography (LC–MS) as the frontrunner in the analysis of processed meat composition in contrast to the ineffectiveness of traditional food analysis methods [48].

Chromatographic techniques also played an important role in the advancements verified in agricultural sciences. The identification and monitoring of biotic and abiotic stress markers [49], and the definition of the state of maturation of certain products [50], help producers define harvest dates according to the intended maturation indicators and considering the organoleptic characteristics of the final product.

In environmental sciences, chromatographic systems also significantly contributed to the search for a cleaner and more sustainable environment. Many environmental pollutants used in agriculture including DDT, lindane, endrin, dieldrin, heptachlor, and chlordane [51], have long biological half-lives and tend to bioaccumulate, therefore, representing a serious threat to different forms of life in our planet, including ours. The identification of such agri-toxics was made thanks to chromatography (GC–ECD, GC–MS and LC–MS, depending on the target analytes). The progressive increase in the sensitivity and precision of chromatographic techniques facilitated the action of governmental agencies, industrial organizations, and research groups on the implementation of inspection guidelines and screening programs that are simultaneously beneficial to air, soil, water, and wildlife as well as to humanity.

In the chemical industry, air monitoring is used to identify and analyse different chemical compounds [52]. Chromatography also plays an important role in the synthesis of radiolabeled chemicals, such as ^14^C-labeled compounds used as radiotracers in metabolism investigations [53].

In pharmacology and biotechnology, chromatography helped in several innovations and advances. From quality control to drug development, and from pharmacokinetics to pharmacodynamics [54], the pharmaceutical industry is heavily regulated to ensure the safety and efficacy of pharmaceutical products and meet the specific guidelines implemented by regulatory authorities worldwide. Additionally, the OMICs platforms allow us to increase our knowledge of the human body at the cellular level since the metabolome offers a powerful way to determine and evaluate the cell response to external and internal stimuli. This understanding is also important to study the impact of drugs and therapeutic procedures at the cellular level. Figure 5 shows a network revealing the interaction and importance of chromatography in the most diverse fields of science, knowledge, and technology, evidencing its impact on those areas.

The growing importance of chromatographic systems in hospitals, research laboratories, and industries expressed, for instance, by the high number of publications (approximately 90,000 papers published in chromatography in the last three and a half years), reveals the enormous contribution of the technique to the advancement of science in various branches of knowledge. Furthermore, its impact in helping the societal and technological challenges posed mainly by climate changes and the COVID-19 pandemic, is outstanding. This contribution was also boosted by the development of other branches of separation science and analytical techniques, namely, MS and highly sensitive detection systems, which developed concomitantly. Overall, the progression of R&D and improvements in chromatographic techniques will continuously improve the accuracy and precision of analytical results, favoring growth and innovation opportunities. The aforementioned fields, among others, boosted several innovations and advancements in the chromatography equipment to improve the technique’s sensitivity and resolution, allowing a wide range of coverage in terms of chemical nature and concentration of the target analytes. Additionally, the developments in column stationary phases favored the improvement of resolution power and increased the number of detectable metabolites. This is exemplified by the huge power of multidimensional chromatography techniques, such as two-dimensional gas chromatography (2D-GC commonly known as GC × GC) [55], and two-dimensional liquid chromatography (2D-LC) [56], for separating complex mixtures.

The scientific world has demanded better performance from chromatographic methods, both in terms of throughput and resolution. On the other hand, the high-resolution power and ultra-high performance, in terms of sensitivity, detection, and identification abilities served as a platform for the evolution of science in a wide range of areas.

### 2.1. Chromatography Helping in Societal Challenges

In this final section, a set of scientific articles published in the last decade was selected to show how the data obtained from the chromatographic analysis can be essential to respond to current societal challenges, with a huge impact on our health and well-being, as well as on the valorization of natural resources and sustainability. Specific case studies are presented to highlight the importance of selecting appropriate chromatographic methods and equipment for the analyses of different types of analytes.

#### 2.1.1. Chromatography-Based Metabolomics Utility to Unveil Disease and/or Physiological Status

The study of a disease, as well as its diagnostics, treatments (including drug development), and/or biomarkers is a very challenging subject matter, and their decoding may be outpaced using chromatographic tools. Indeed, health-related studies based on the human metabolome are quite challenging. The human metabolome is very complex, and the use of animal models may be required to help in a wide range of scientific queries, from basic science to the development of new therapies and vaccines, among others. Indeed, the human metabolome contains a vast chemical diversity of the low-molecular-weight substances that can be produced in cells along a metabolic process, which include amino acids, carbohydrates, lipids, nucleic acids, vitamins, organic acids, and volatile metabolites, among others. Moreover, each body fluid or cell type may present a distinctive metabolite profile (also considering a wide concentration range), which can depend on genetic and environmental factors [57]. As there is no analytical technique that can cover all the enumerated specificities, the combination of several analytical techniques may be crucial to have an in-depth metabolome characterization. One example was shown in the multi-omics approach that was performed to clarify the molecular connection between obesity development and the high-fat diet, using the mouse model C57BL/6J [58]. For this study, data RNA transcriptomics of tissues, the metabolomics of several body fluids (plasma and urine) and tissues (liver, adipose, and muscle), and metagenomics analysis were combined to achieve the goal. Hence, the mice metabolome was studied using diverse analytical techniques, namely: NMR (Nuclear Magnetic Resonance), which allows the analysis of non-polar and polar analytes, is highly reproducible, but has lower sensitivity; LC–MS has higher sensitivity and depending on the LC type can detect a broad range of components, and in this case, a UPLC-qToFMS and UPLC-Q extractive high-resolution accurate mass (HRAM) were used, and the instrumental conditions were adjusted according to the sample type (e.g., different chromatographic columns—SeQuant ZIC-cHILIC, BEH C18, and HSS T3 (with <3 µm particle size), different mobile phases, etc.); and GC–MS, which is also a very sensitive and reproducible technique, and in this case, analytes were derivatized by silylation and analyzed by a GC-ToFMS, equipped with an RTX 5 column. The MS analyzers used in this study, namely, qToFMS and HRAM, are highly selective and sensitive, being extremely important for accurate metabolite identification and quantitation. The metabolites profile obtained from mice tissues and body fluids were combined and analyzed by a multi-block PCA (Principal Component Analysis) approach (Figure 6), which showed a distinguished pattern regarding the type of diet: low-fat (LFD) and high fat (HFD). LC–MS urine profile was the most distinctive profile among these samples, and there were also alterations observed in the liver tricarboxylic acid (TCA) cycle intermediates concentrations (e.g., malate, fumarate, oxaloacetate), which corroborated the gene expression data obtained of the enzymes related to TCA cycle [58].

Reliable interpretations and, therefore, biological data mining of the huge amount of data that can be acquired by the chromatographic techniques, particularly the multidimensional ones, are vital, especially in health-related studies. In fact, omics data can be processed using several free available bioinformatics tools that integrate and correlate the data with several databases [59], as previously shown with multi-block PCA. However, for deeper knowledge, omics data can be fused and integrated using artificial intelligence and machine learning approaches, which combine and process chromatographic data to help researchers and physicians to deepen understanding and create knowledge that supports their decisions [60,61,62]. This remarkable amount of data that is continuously been produced every day has been contributing to the construction of online databases, which systematize and may integrate several data domains, depending on the database [61].

One innovative and disruptive example that combines chromatography and artificial intelligence and machine learning approaches, in a clinical context, was a fast GC methodology that was developed for a possible non-invasive diagnosis of Parkinson’s Disease (PD) using the odor profile of humans’ skin sebum [63]. The developed artificial intelligence olfactory system consisted of three modules: an injection and preconcentration module (adsorbent tube filled with Tenax TA), a GC module (DB-1 column—1m), and a surface acoustic wave sensor detection module (Rayleigh wave gas sensor with a −69,766 Hz/ng sensitivity to mass deposition), with embedded machine learning (ML) algorithms. The linear detection range of the methodology varied from 0.025 to 50 mM. Three volatile organic compounds (VOCs) showed a significantly different profile between the PD patients and healthy controls, for instance, octanal, hexyl acetate, and perillic aldehyde. Depending on the ML algorithm, the models could achieve up to 92% regarding specificity and sensitivity. An accuracy of 70.8% was achieved for the classification model that was based on the significant features. This newly developed system has several advantages, namely, it is non-invasive, user friendly, convenient, and fast, which allows it to be easily implemented in hospitals and/or clinics to diagnose and monitor PD treatments. Still, several limitations need to be overcome, namely, co-elutions may occur, which may compromise the results; the diagnostic accuracy is still dependent on the training set dimension and population representativeness of the samples, which promotes a limited model utility for now; additionally, the biological relevance of hexyl acetate and perillic aldehyde is still not well-understood, which may require the addition of new biomarkers [63].

The complexity of the body fluids and tissue matrices, or even the detection of trace components require the use of highly sensitive and throughput methodologies, such as multidimensional chromatography. Comprehensive two-dimensional gas chromatography coupled to time-of-flight mass spectrometry (GC × GC–ToFMS) is a well-suited technique to perform personalized studies focused on individuals, capable to detect analytes at pg level, as it can be observed on the study of the usage impact of surgical face masks on young researchers (individual protection that became daily habit due to the COVID-19 disease), taking into consideration a normal working day [64]. For this study, exhaled breath condensate was the selected body fluid for the analysis, and its volatiles were extracted by SPME. A non-polar stationary phase (HP-5, 30 m) was used as the first column, which was combined with a polar stationary phase (DB-FFAP, 0.79 m) that was used as the second column, with these two columns being connected in series by a cryomodulator (highly recommended for the analysis of highly volatile metabolites). The high acquisition speed (ca. hundred spectra per second) makes ToFMS the most suitable analyzer for GC × GC once it provides enough data density, thus, allowing the accurate spectra deconvolution of overlapping peaks. The lipid peroxidation volatile profile achieved by GC × GC–ToFMS did not show significant changes along a working day, namely, aliphatic alkanes and aldehydes (which were previously associated with the lipid peroxidation process [65]). Additionally, heart rate and blood oxygen saturation count did not display significant changes, and the reported values were within the conventional values predictable for a healthy adult. Thus, in the tested conditions, the use of face masks did not promote significant alterations in pulmonary hemodynamics, which might lead to hypoxia and resultant lipid peroxidation [64].

#### 2.1.2. The Role of Chromatography to Respond Current Agri-Food Industries Challenges and Sustainability

Food consumption trends have been changing for years, and currently, consumers are looking for healthier products, not only in nutritional terms, but also considering food safety standards and the presence of bioactive components with beneficial properties (e.g., functional foods, nutraceuticals, etc.) that are convenient, natural, and sustainable. Consumers are looking for local and organic products [66,67] presenting a lower ecological footprint, and fewer agrochemical treatments during their production, among others. To meet these requirements, food producers need to continuously develop new food products, and decisive and accurate analytical techniques are required to monitor food authentication and traceability, both ensuring food quality [68]. Moreover, the exaggerated competition in the production of certain food products has been promoting food fraud/adulteration, and chromatographic techniques play an important role in their monitoring. Another emerging topic is the amount of food loss and waste and how it should be reused. Indeed, ca. 90 million tons of food are wasted by the European Union each year [69], leading to significant environmental and economic issues, and food industries are one of the players involved. Therefore, there is an active search, for instance, using chromatographic techniques for new bioactive components and/or new functional food ingredients using agri-food industrial wastes or by-products in a wide range of applications, as it is already reported in several recent reviews [70,71,72,73].

Food matrices are complex due to the wide range of chemical components that they contain (e.g., different polarities, structures, concentration levels—from ng/kg to g/kg), which chemical analysis implies a selection of the proper analytical technique may fulfil the specific requirements of its analysis (target or untargeted analysis). For instance, GC-based methodologies showed to be a very valuable analytical technique to monitor food VOCs that may have an impact on their aroma if above their odor threshold [74]. Indeed, VOCs have been studied to allow the food’s distinction regarding its geographical region, species or varieties, using mainly HS-SPME as the extraction technique, as shown in the following examples: apple (fruit and juice) and cider [75], broa (Portuguese maize bread) [76], grape [77,78], onion [79], saffron [40], olive oil [80], salt [81], rice [82,83], truffles [84], pears [85], among others. Considering the detection technique, gas chromatography-quadrupole mass spectrometry (GC-qMS), is the most used [40,75,77,79] that can be easily implemented in an industrial context. GC × GC–ToFMS, however, has also been used [76,78,81,85] due to lower detection limits, faster run times, and high resolution and peak capacity, promoting an in-depth food characterization. Nevertheless, GC × GC-based techniques have several limitations, such as high consumable costs, complex instrumentation, and the requirement of operational expertise. Since the type of analytes that are mainly present in these food matrices have polar groups in their chemical structure (e.g., acids, alcohols, aldehydes, esters, furans, ketones, lactones, norisoprenoids, pyrazines, terpenic compounds, etc.), polar stationary phases are preferable for this analysis, namely, polyethylene glycol derivative phases (e.g., BP-20, SUPELCOWAX 10, and DB-FFAP) [40,75,77,79]. For GC × GC–ToFMS analysis, the combination of non-polar/polar columns has been used, for instance (5%-phenyl)-methylpolysiloxane phase as the nonpolar columns (e.g., Equity-5, HP-5) and polyethylene glycol derivative phase (e.g., DB-FFAP) as the polar columns [76,78,81].

Enantioselective GC-based approaches are required for the analysis of chiral volatile components to assess the authenticity and quality of foods, and, for instance, the enantiomeric ratio has been used for the origin identification and adulteration of essential oils [86,87] and for the chiral contribution on baked/fermented teas [88,89], wines [90,91], among others. Particularly, in recent years, enantioselective multidimensional GC (enantio-MDGC) has been a key technique used for the separation of overlapping enantiomers, in a wide range of foods [92]. For instance, tea tree oil (TTO), extracted from *Melaleuca alternifolia*, is one of the most commercialized essential oils worldwide, whose demand is higher than its production, leading to intentional adulterations. The normative ISO 4730, which regulates the TTO chemical composition, has been continuously updated due to the outcomes achieved by the scientific research, which contributed to stricter requirements of the chromatographic profile at each update [93]. Enantioselective gas chromatography (eGC), particularly fast chiral methodologies based on GC–eGC-FID and eGC × GC-FID were applied to study the stereoisomeric ratios of limonene, terpinen-4-ol and terpineol in TTO [94]. The heart/cut approach (GC–eGC-FID) was achieved using a microfluidic Deans switch apparatus, with the Rxi-17 Sil MS used as the first-dimension column and Astec CHIRALDEX B-PM used as the second-dimension column; while the comprehensive approach (eGC × GC-FID) was realized with Astec CHIRALDEX B-PM as the first dimension column and SUPELCOWAX 10 as the second-dimension column. Both approaches used cryogenic modulators, ideal for volatile components. An example of the eGC × GC-FID contour plot of two TTO can be observed in Figure 7, in which it is possible to verify the adequate enantiomer separation of limonene, terpinen-4-ol, and terpineol without interferents [94]. The purposed chromatographic methodologies allowed a significant reduction in run time, from 75 min (ISO 4730, GC-FID) to 25 and less than 20 min for GC–eGC-FID and eGC × GC-FID, respectively [94]. Therefore, these fast and accurate approaches may act as trustworthy platforms for authenticity control of TTO [94], and they contributed to the amendment of ISO 4730:2017, which added the enantiomeric distribution of terpinen-4-ol that TTO must have in its composition, for instance: 67–71% (+, S) and 29–33% (−, R) [95].

LC-HRMS can also be used to unveil the complexity of the food matrices and the great variability of the bioactive components, as this emerging technique guarantees the unambiguous determination of the elemental content of isobaric compounds and resolves the co-elution problems of isobaric compounds due to the type of analyzer, with ToF and Orbitrap the most frequently used [96]. One example is the use of a LC-QToFMS methodology to study saffron adulterations, which have been occurring with foreign plants that present similar color and morphology (the normative ISO 3632 evaluates and determines the saffron quality, of which detection limit of adulteration is up to 20%). Particularly, saffron adulteration has been performed with gardenia (*Gardenia jasminoides Ellis* of the *Rubiaceae* family), whose major glycoside is geniposide. For this purpose, the better peak capacity and resolution of the LC-QToFMS methodology were achieved with the Ascentis Express Fused-core C18 column (fused-core particles composed of 2.7 µm particle size and 0.5 µm thick porous shell). This methodology was shown to be useful once it was able to simultaneously quantify glycosylated kaempferol derivatives, which were considered saffron authenticity markers, and geniposide as an adulteration product in saffron [97]. Depending on the glycosylated kaempferol that was used, the minimum detectable adulteration of saffron varied between 0.2 and 2.5%, a fact that is significantly lower than the 20% of the ISO 3632.

The combination of several chromatographic techniques may be applied to a broader strategy for the multisensorial perception of foods due to the study of several dimensions associated with aroma perception, as was shown in the study of coffee odor and taste [98]. Indeed, there is a multimodal perception along with coffee or other food consumption, i.e., stimuli generate simultaneous (or closely simultaneous) data in more than one sensory modality. Therefore, the coffee multimodal perception was studied using the fingerprints of the volatile and non-volatile fractions and combined with sensory data [44]. The volatile profiles were obtained using HS-SPME combined with GC-qMS (equipped with a SolGelwax column). The non-volatile profiles were achieved using an LC-UV/DAD (equipped with a Platinum EPS C18 column), and analytes identification was confirmed using an LC–MS/MS with a triple quadrupole (equipped with an Ascentis Express C18 column). The three data domains were processed using unsupervised (PCA and MFA—multifactor authentication) and supervised (PLS—Partial Least Squares Regression) chemometrics tools. The developed regression models showed a key role of the volatiles in the sensorial characterization of the samples, with it being hypothesized that VOCs profile may be satisfactorily representative to define coffee flavor. Nevertheless, an integrated approach should be well-suited as a complement to sensory analysis, giving particular interest in the designing of new coffees (e.g., with different flavor profile). The main limitation of this approach is the consistency of the sensory data due to the subjects’ subjectivity. These sensory prediction instrumental tools will require the acquisition of a significant amount of chemical and sensorial data in order to be more robust and reliable, and the use of artificial intelligence and machine learning algorithms will become useful tools to combine data for the understanding of such complex phenomenon as it is the multimodal flavor perception [98].

The in-depth chemical characterization of food wastes and by-products, as potential sources of high-added value compounds, may presuppose the design of an analytical strategy that includes a set of equipment according to the nature and concentration of the target analytes. For instance, the lipophilic components can be derivatized by silylation and analyzed by GC-MS [99,100,101,102], and the phenolic components can be analyzed using LC–MS [99,103,104]. Indeed, the integration of several analytical methodologies allows for a more comprehensive and detailed chemical characterization as was shown in the study of the potential bioactive components from *Passiflora mollissima* seeds [99]. A pressurized-liquid extraction was optimized to be applied to the seeds, in which the non-polar extract was derivatized by silylation and analyzed by GC-QToFMS (equipped with a DB-5 column); while the polar extract extracted from the defatted residue was analyzed by UHPLC-qToFMS/MS (equipped with a Zorbax Eclipse Plus C18 column), as observed in the workflow presented in Figure 8.

The semi-targeted GC–QToFMS and untargeted UHPLC–qToFMS/MS strategies were successfully implemented to unravel the composition of *P. mollisima* seeds, particularly the polyphenols-rich extract, which was composed of flavonoids, flavanols, and proanthocyanidins, while the oily fraction was composed by MUFAs and PUFAs [99]. This strategy is particularly useful for the bioprospection of agri-food wastes and by-products, as it is extremely important to have high-resolution MS and MS in tandem (as it was used in this previous study, namely, QToFMS and qToFMS/MS) to perform putative identification of analytes, as their accurate identification is only proven through the authentic standards co-injection, and most of them are not commercially available or can be economically expensive. Nevertheless, these types of equipment require operational expertise and are quite expensive, which makes them still far from part of routine laboratory analysis, with quite a margin for improvement. Therefore, equipment such as HPLC-DAD/FLD can be used as a cost-effective option, particularly when authentic standards are available and there is previous knowledge regarding sample composition [105]. The use of libraries, as well as databases, will help in the putative identification of the new detected components. The comprehensive and detailed characterization of such complex natural matrices is crucial for the further steps of investigation, namely, regarding the bioprospection and consequent valorization of the food wastes and by-products as the potential source of bioactive compounds, that may be further used as food or cosmetic components.

#### 2.1.3. Using Chromatography to Monitor Environmental Contaminants

The ecosystems are unfortunately widely contaminated with several ranges of chemicals, which according to European legislation, are defined as “substances (i.e., chemical elements and compounds) or groups of substances that are toxic, persistent and liable to bio-accumulate and other substances or groups of substances which give rise to an equivalent level of concern” [106]. Within these substances, included is microplastics and emergent pollutants (such as sucralose and other artificial sweeteners; nanomaterials; per- and polyfluoroalkyl substances; pharmaceutical and hormones; drinking water and swimming pool disinfection byproducts; sunscreens/UV filters; brominated and emerging flame retardants; dioxane; naphthenic acids; benzotriazoles and benzothiazoles; algal toxins; ionic liquids; halogenated methanesulfonic acid). Therefore, it is important to establish mitigation strategies to monitor and reduce these contaminations that may be present at trace levels (from parts-per-trillion, ppt, ng/L, to parts-per-billion, ppb, μg/L), and chromatographic techniques may be helpful in achieving this intention.

Microplastics (MP) contamination is not only restricted to marine ecosystems as it also affects the human food system, and their identification and quantification can be performed through visual recognition, spectroscopic analysis (e.g., FTIR), tagging methods, and chromatographic techniques. Indeed, sequential pyrolysis GC-MSD (mass selective detector), equipped with an HP-5 column, showed to be an appropriate tool for the identification of types of polymer of MP particles and associated organic plastic additives [107], particularly plasticizers (e.g., phthalates), flavoring agents (e.g., benzaldehyde), and antioxidants (e.g., 2,4-di-*tert*-butylphenol) in MP particles. The characteristic decomposition products allowed for the identification of isolated MP particles, comparing them with standards [107]. The main disadvantages of this method rely on the fact that only one particle can be analyzed per run, being non-appropriate for routine analysis; additionally, it is not adequate for the study of entire environmental samples, as a matrix effect occurs and it is very sensitive to contaminations. Another thermoanalytical method that combines thermogravimetric analysis coupled to solid-phase extraction (TGA–SPE) and thermal desorption GC–MSD (using an HP-1 MS column) was used to quantify polyethylene in several environmental samples (e.g., soil, suspended solids, or bivalves) [108]. This methodology showed to be robust, however, still requires further studies with a broader range of MP. The potential of chromatography has yet to be fully exploited, nevertheless, these thermoanalytical methods will allow the first sample screening to determine contamination levels, with posteriorly applied spectroscopic methods being used to obtain the precise determination of the number and size of the particles [109].

LC–MS/MS (with triple quadrupole) has been used to quantify MP, namely, polyethylene terephthalate (PET) and polycarbonate (PC) in the dust (important carrier of several contaminants), using alkali-assisted thermal depolymerization. The advantage of using LC–MS/MS methodology to quantify atmospheric MP is that there is no need for pretreatments of the samples, once it can be directly used, also a wide range of MP sizes can be analyzed [110,111,112]. For instance, the minimum limit of quantification (LOQ) of PET and PC, using LC–MS/MS, was 178.3 μg kg^−1^ and 27.7 μg kg^−1^, respectively [110], the values of which were significantly lower than those using an LC-UV (1 × 10^6^ μg kg^−1^ for PET) [113]. Despite the better sensitivity of LC–MS/MS, it is not routine laboratory equipment, requires qualified people to operate it, and is expensive.

Adsorptive microextraction techniques coupled to HPLC-DAD have been used to quantify contaminants in several types of water (e.g., seawater, wastewater, superficial water) from several sources, for instance: phenols that are intermediates in industrial processes (e.g., production of pesticides, explosives, pharmaceuticals, resins, among others) [114]; benzotriazoles (usually used as anti-corrosion additives, ultraviolet stabilizers, fungicides, or dyes), benzothiazoles (generally used as anti-corrosion additives, paper biocides, or herbicides and fungicides), and benzenesulfonamides (largely used as plasticizers, intermediate in sweeteners synthesis, or disinfectants) [115]; or anthropogenic drugs, such as caffeine and acetaminophen [116]. Despite the high selectivity of the adsorptive microextraction techniques and the requirement of low sample volume, the detection limits provided by HPLC-DAD may not be satisfactory. For instance, bisphenol-A (BPA) was quantified by BAµE-LD/HPLC-DAD (equipped with a Tracer excel 120 ODS-A column), with a detection limit of 0.3 μg L^−1^ [114], of which the value is significantly higher than reported concentrations of BPA in US waters (for instance 0.0009 μg/L) [117].

GC–MS/MS and LC–MS/MS were used to perform an ultra-trace analysis of 40 French bottled mineral glasses of water, in which a broad range of emergent contaminants was quantified, for instance, 118 pesticides and respective transformation products, 8 alkylphenols, 172 pharmaceuticals, 11 phthalates, 11 hormones, and 10 perfluoroalkyl substances (PFAS) [118]. Considering the nature of the contaminants, several analytical producers were applied to these mineral waters, namely: offline SPE LC–MS/MS (pharmaceuticals, hormones, pesticides, PFAS, and alkylphenols), online SPE LC–MS/MS (pharmaceuticals and pesticides), Stir Bar Sorptive Extraction (SBSE) GC–MS/MS (pesticides) and online SPME GC–MS/MS (pesticides, alkylphenols, and phthalates). These previous approaches were optimized to provide the lowest and most reliable limit of quantification (LOQ), without matrix interferents. Most of the LOQ were below 10 ng/L, and almost all contaminants (99.7%) registered concentrations lower than LOQ [118]. These waters did not present hormones and pharmaceuticals in their composition, which indicates the protection of springs utilized for bottling from contamination. However, some samples presented pesticides and their metabolites, PFASs, and alkylphenols in their composition at low levels (two times lower than French regulation of pesticides), which suggests the possible use of herbicides near the aquifers (surface water) and distant atmospheric transport and deposition (e.g., by rainfall). The importance of chromatography was well underlain by this previous example: on the one side, the developed tool helps the water industry in the monitoring contaminants at ultra-trace concentrations, and on the other side, the trustworthy measurements avoid false positive/negative results.

The structural and molecular characterization of water-soluble organic components from air particles can be studied by different analytical methodologies, and the most pertinent are systematized in Figure 9, considering their resolution with the mass of the water-soluble organic matter (WSOM) analyzed [119]. In fact, WSOM plays a key role at environmental, climate, and public health levels, nevertheless, the knowledge regarding their sources, composition, formation, and transformation mechanisms is still scarce, being a current topic of research. Online aerosol mass spectrometry (AMS) is the preferred technique to analyze WSOM, nevertheless, it cannot overcome the high-resolution achieved in multidimensional offline techniques. Indeed, great enhancement in the WSOM constituents’ selectivity and molecular discrimination has been reached with the utilization of HR-MS detectors and their hyphenation with multidimensional chromatographic equipment, or in combination with EEM–PARAFAC (fluorescence excitation-emission matrices–parallel factor analysis) methods. Moreover, 1D NMR (mainly liquid-state) has also been used for the structural characterization of aerosol WSOM, and multidimensional (2D or higher) liquid-state NMR arises as a very promising technique for this purpose. Even so, 2D NMR still faces several limitations, such as the handling difficulties regarding the processing of overlapping data, the lack of analytical knowledge in the utilization of spectral editing techniques of NMR, and the requirement of chemometric tools that can handle big data sets, such as those generated by multidimensional approaches [119].

## 3. Final Remarks

The ever-increasing need for low-level, accurate determination of several kinds of analytes in samples from different environments, has been satisfied by remarkable advances in highly sensitive chromatographic approaches, and the associated detection and data-handling systems. These advances have facilitated the measurement of analytes in complex matrices at increasingly lower levels. Never has the demand for chromatographic information been as fundamental as now for assisting in important societal challenges that humanity is facing along with the life sciences regarding the environment, and from climate change to energetic security, where the increasing use of renewable energy sources towards a decarbonized society is pivotal for planet sustainability, based on the best scientific knowledge available. From a sustainable point of view, the down-scaling of chromatography is expected since it will reduce solvent consumption and consequently lessens the environmental impact and solvent storage and disposal costs. The increasing use of chromatographic techniques in recent years, expressed in almost 90 thousand published papers from 2019 to the middle of June 2022, and with 53.7% of publications in chromatography from the last 80 years conducted in the last 20 years (PubMed, 26 June 2022), it is expected that this powerful analytical tool will continue to develop over the next few years, opening new avenues for scientific advancements and helping societal alleviate challenges, allowing to capitalize on research as a tool to solve problems. Therefore, the context itself is pushing new paradigms, and setting new challenges for the future development of chromatography. In this regard, fields such as the life sciences, bioanalysis, pharmaceutical and clinical analysis, environmental sciences, agricultural sciences, food science and, more recently, OMICs, are demanding higher coverage of analytes at trace levels. In response to the emerging global challenges, it is of utmost importance that worldwide companies and academic laboratories offer to look for solutions through joint initiatives to have a greater impact on social, economic, and environmental dimensions. Chromatography through several fields of science will help to find solutions to maintain the integrity of the life support systems on a global scale, addressing a wide range of subjects, such as biodiversity loss, pollution, land, and water degradation, as well as climate change. In this context, on the one hand, GC–eGC-FID, GC–MS, LC–MS, GC–QToFMS, GC × GC–ToFMS, UHPLC–qToFMS/MS, among others, emerge as techniques that highly improve sensitivity, resolution power, and identification capacity. On the other hand, the development of novel capillary column materials with higher surface area, alternative geometries, longer lifespans, and balanced polarity coverage is also encouraged.

## Figures and Tables

**Figure 1 molecules-27-05267-f001:**
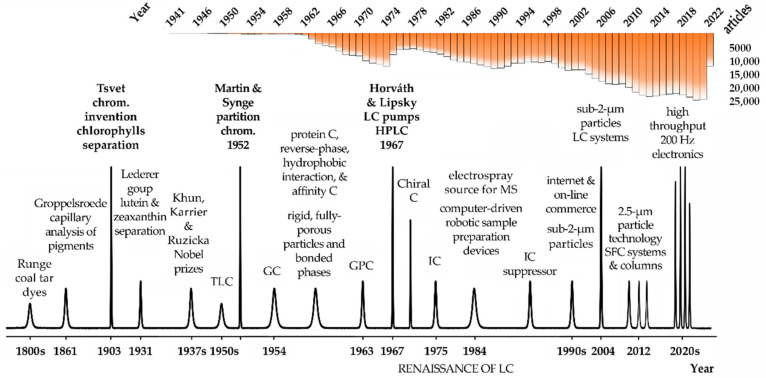
Overview of major milestones in the evolution of chromatography. The number of articles published in the literature according to PubMed servers and using the keyword chromatography are indicated in the top part of the figure. Legend: C—chromatography, GC—gas chromatography, GPC—gel permeation chromatography, HPLC—high-performance liquid chromatography, IC—ionic chromatography, LC—liquid chromatography, MS—mass spectrometry, SFC—supercritical fluid chromatography, TLC—thin-layer chromatography.

**Figure 2 molecules-27-05267-f002:**
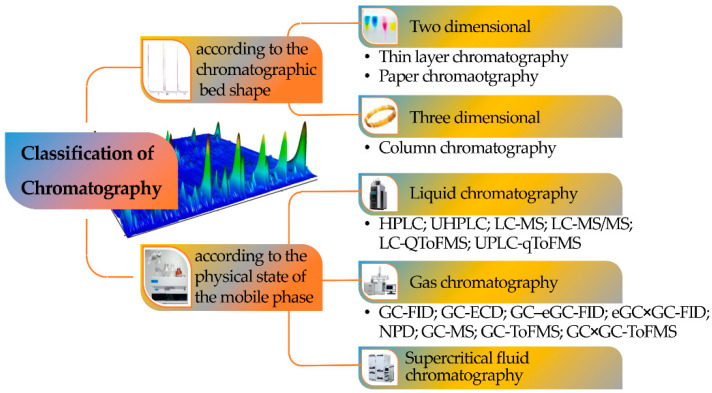
Classification of chromatography according to the chromatographic bed shape and the physical state of the mobile phase.

**Figure 3 molecules-27-05267-f003:**
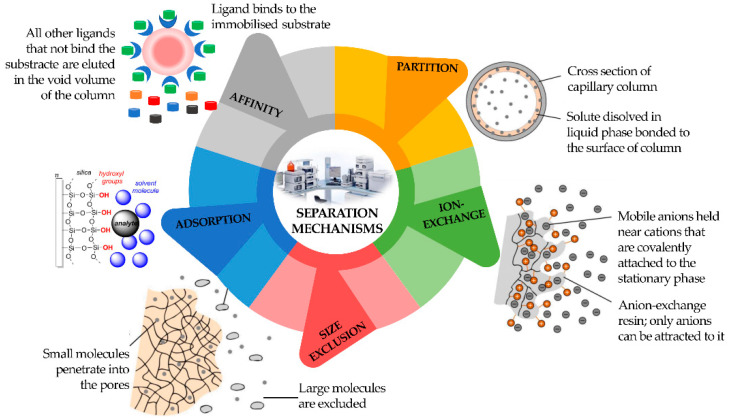
Classification of different types of chromatography according to separation mechanism.

**Figure 4 molecules-27-05267-f004:**
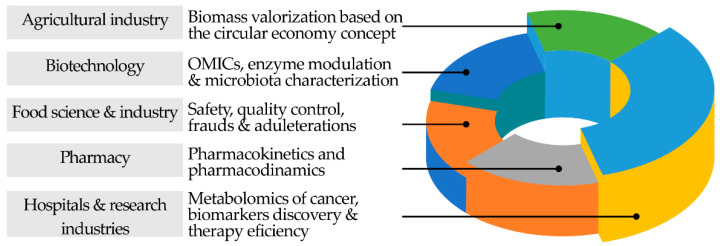
Important applications fields of chromatography in science development.

**Figure 5 molecules-27-05267-f005:**
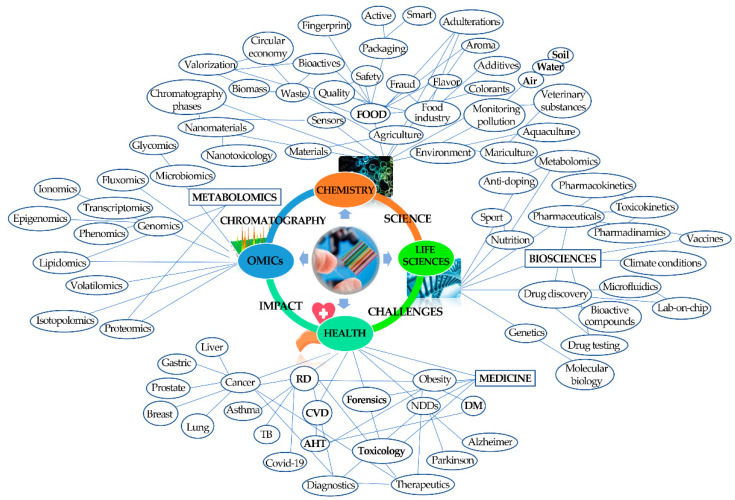
Co-occurrence network map related to the importance of chromatography in some fields of science.

**Figure 6 molecules-27-05267-f006:**
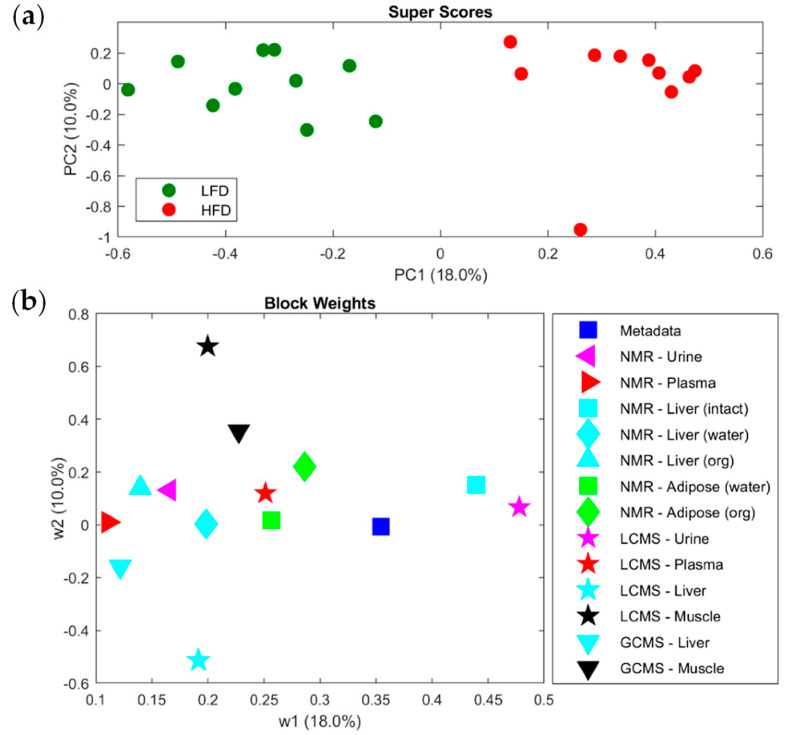
Multi-block PCA super scores plot (**a**) and block weights (**b**) of the body fluids and tissues, considering the metabolomics data obtained from mice fed with a low-fat diet (LFD) and high-fat diet (HFD) [58].

**Figure 7 molecules-27-05267-f007:**
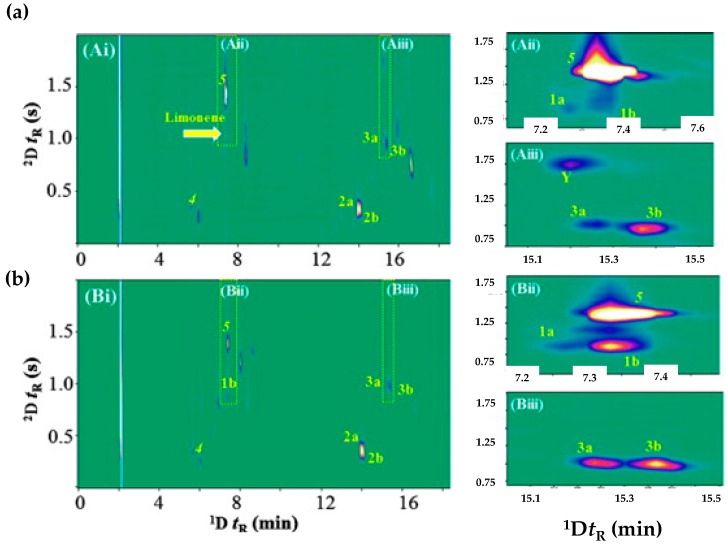
eGC × GC-FID contour plots of two TTO samples P1 (**a**) and C6 (**b**), and the expansion of rectangle region of limonene (Aii and Bii) and α-terpineol (Aiii and Biii) in TTO of samples P1 and C6, respectively. Components: 1(a), (−)-limonene; 1(b), (+)-limonene; 2(a), (+)-terpinen-4-ol; 2(b), (−)-terpinen-4-ol; 3(a), (−)-α-terpineol; 3(b), (+)-α-terpineol; 4, α-pinene; 5, p-cymene; Y, unknown compound. Reprinted from [94].

**Figure 8 molecules-27-05267-f008:**
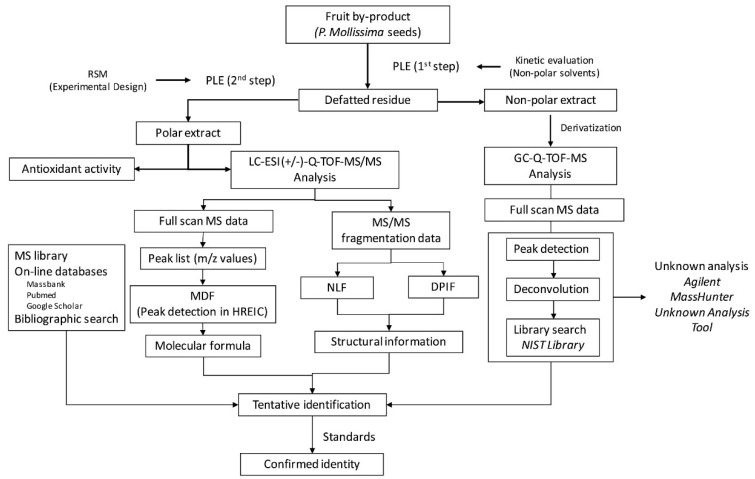
Workflow of the detailed chemical characterization of the bioactive components from *Passiflora mollissima* seeds. Legend: MDF: mass defect filtering; DPIF: diagnostic product ions filtering; HREIC: high-resolution extracted ion chromatograms; NLF: neutral loss filtering; PLE: pressurized liquid extraction; RSM: response surface methodology. Reprinted from [99].

**Figure 9 molecules-27-05267-f009:**
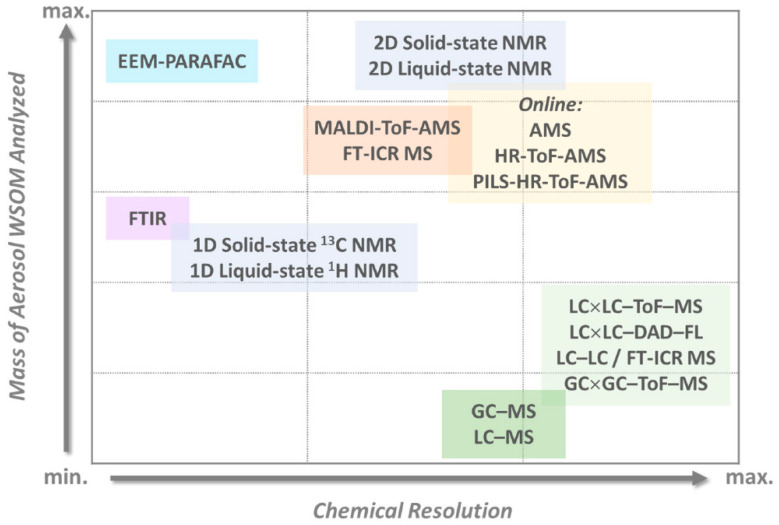
Schematic illustration of the analytical strategies most currently employed in the structural and chemical characterization of aerosol water soluble organic matter (WSOM), considering its amount versus the chemical resolution of the applied methodology [119].

## Data Availability

Not applicable.

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
