# Peer review of "Chromatographic-Based Platforms as New Avenues for Scientific Progress and Sustainability"

_molecules, 2022, doi:10.3390/molecules27165267_

Round 1

Reviewer 1 Report

Referee report for the manuscript molecules-1819870-peer-review-v1

Chromatographic-based platforms: new avenues for science progress and sustainability”

By José S. Câmara*, Cátia Martins, Jorge A. M. Pereira, Rosa Perestrelo, and Sílvia M. Rocha

Attempting to write a review about the last 120 years of history and nearly all current applications of modern chromatography in about 25 manuscript pages and about 100 references is a megalomanic endeavor that is clearly doomed to fail. Unfortunately, this review is no exception.

A review is expected to be focused, factual, critical, and timely to be relevant for the reader, and to my opinion, this manuscript fails in all these criteria.

However, let us start with the good things: I enjoyed reading the first (introductory) part of this manuscript. It is an easy-reading introductory text that I would not hesitate to give to my undergraduate students as preparatory material for their separations science course. It lacks a bit structure and profoundness, but this is still acceptable.

I also have enjoyed very much the beautiful figures: Although their relevance may be sometimes questionable, they provide beautiful visualizations of some interesting aspects in chromatography. Well done !

The rest, and this regrettably starts already in section 2 (l. 392 ff.), I did no longer enjoy.

Instead of being factual, the authors too often make personal, almost political comments which – if at all – can be part of the concluding discussion and evaluation, but not of the review itself.

Also the review is very repetitive in its general statements.

And it lacks a clear structure and system in the organization of the incredible abundance of literature published.

Furthermore, I do strongly disagree with the authors that the incredible increase in publication numbers is a valid proof for the importance of chromatography to solve everyday’s problems. This statement would perhaps be possible if the authors undertook to compare the development of publication numbers in separation science or chromatography with the publication numbers for other analytical techniques, such as mass spectrometry or sensors.

Moreover, I do not share the optimistic belief that the author’s have in the relevance of increased numbers of publications in the last years or decades: It is the old mistake to measure quality by quantity. Yes, the number of publications in the field of separation science has increased – but this is the same situation for all fields of science, not only for chromatography. Moreover, it is remarkable that hardly any journal that is entirely devoted to Separation Science can be found in the Q1 (= first quarter) of the journals in its discipline. This, in my opinion, is due to the fact that these journals have decided to publish applications, rather than real novel fundamental, instrumental or methodological developments which still is the case in other sub-disciplines of analytical chemistry. So, yes, the number of publications has clearly increased, their impact (on average) has not.

Along this line of argumentation, it is not clear according to which criterion the authors have selected the 118 references that are reported in this review (from the 20,000 to 25,000 chromatography-related publications that have appeared every year during at least the last decade).

In any case, quoting (at least) 18 papers form the authors’ own production seems to put a bit too much emphasis on the own work.

Not only the selection of publications, but also the selection of items that are mentioned appears rather arbitrary: Large parts of this review read like “this author has done this and that author has done that”. The presentation of the reviewed material in this way does not represent an additional benefit to the reader as the critical discussion of individual methods’ advantages, limitations and applications is missing.

I have made a number of further – both lingual and scientific – comments directly into the pdf file of the manuscript. However, I cannot recommend publication of this manuscript as it does not present any actual benefit to the reader: Instead, the reader is referred to two wonderful and still valid books on chromatography, one being L.S. Ettre’s “Chapters in the Evolution of Chromatography”, Imperial College Press London, 2008) which very entertainingly describes the history of chromatography, and the other being Colin Poole’s “The Essence of Chromatography” (Elsevier, 20

Author Response

REVIEWER #1

Reviewer#1 Attempting to write a review about the last 120 years of history and nearly all current applications of modern chromatography in about 25 manuscript pages and about 100 references is a megalomanic endeavor that is clearly doomed to fail. Unfortunately, this review is no exception.

A review is expected to be focused, factual, critical, and timely to be relevant for the reader, and to my opinion, this manuscript fails in all these criteria.

AUTHORS ANSWER: We appreciate and respect the reviewer's comments and opinions. We agree that writing a review about the last 120 years of history and nearly all current applications is not an easy task. However, that was not our aim. We intended to point out some of the milestones that determined the development of chromatography and how this path became determinant in the evolution of science and society. Chromatography became a technique of great importance and wide range of applicability and so it is impossible to include all aspects in a review manuscript. To do that we would need a few books. So, as referred, we mainly intend to draw attention to the importance that chromatography had and still has for the evolution of science and society, highlighting the points that we consider to be of interest to the readers of Molecules. Despite this journal is not specific to the chromatographic field, it covers several areas of the scientific knowledge in which chromatography became an essential tool and so we believe it will be of interest to a vast audience of Molecules. In order to increase the manuscript focus the section 2 was significantly reduced and the section 3 was included in section 2.

Reviewer#1 However, let us start with the good things: I enjoyed reading the first (introductory) part of this manuscript. It is an easy-reading introductory text that I would not hesitate to give to my undergraduate students as preparatory material for their separations science course. It lacks a bit structure and profoundness, but this is still acceptable.

I also have enjoyed very much the beautiful figures: Although their relevance may be sometimes questionable, they provide beautiful visualizations of some interesting aspects in chromatography. Well done !

AUTHORS ANSWER: We appreciate the reviewer's comment and opinion very much. Evidently that some fundamentals of the technique were considered in this review, including the historical evolution, which will be of great interest to those who are introduced to the world of chromatography. For those who are already experts in the field, it is always good to review some aspects of chromatography that go unnoticed daily. Eventually, these readers would appreciate a little more deepness in some subjects, but it is impossible to attain that without making the review too long and unpractical to read.

Reviewer#1 The rest, and this regrettably starts already in section 2 (l. 392 ff.), I did no longer enjoy. Instead of being factual, the authors too often make personal, almost political comments which – if at all – can be part of the concluding discussion and evaluation, but not of the review itself. Also the review is very repetitive in its general statements.

AUTHORS ANSWER: We thank the reviewer for the comment. In fact, section 2, referring to the Contributions of chromatography to science progress, allows for the inclusion of several personal opinions supported by the authors' knowledge and supported by resources from published works. That's what we tried to do and convey. Nevertheless, we agree that it would be important to make the content more comprehensive and universal and less personal and politicized and so the MS was revised accordingly.

Reviewer#1 Furthermore, I do strongly disagree with the authors that the incredible increase in publication numbers is a valid proof for the importance of chromatography to solve everyday’s problems. This statement would perhaps be possible if the authors undertook to compare the development of publication numbers in separation science or chromatography with the publication numbers for other analytical techniques, such as mass spectrometry or sensors.

AUTHORS ANSWER: We appreciate the comment. We agree that the original text sounds a little biased toward the role of chromatography in the evolution of science as a sole contributor. Based on the reviewer's suggestion, we revised the MS to highlight other branches of science that, together with chromatography, gave an important contribution to the growth and evolution of science and contributed to the resolution of many societal challenges. In fact, the evolution of the detection architectures used in chromatography and other hyphenated techniques were extremely important to drive this evolution.

Reviewer#1 Moreover, I do not share the optimistic belief that the author’s have in the relevance of increased numbers of publications in the last years or decades: It is the old mistake to measure quality by quantity. Yes, the number of publications in the field of separation science has increased – but this is the same situation for all fields of science, not only for chromatography. Moreover, it is remarkable that hardly any journal that is entirely devoted to Separation Science can be found in the Q1 (= first quarter) of the journals in its discipline. This, in my opinion, is due to the fact that these journals have decided to publish applications, rather than real novel fundamental, instrumental or methodological developments which still is the case in other sub-disciplines of analytical chemistry. So, yes, the number of publications has clearly increased, their impact (on average) has not.

AUTHORS ANSWER: Yes, we agree with the reviewer that the number of publications increased overall science fields, and we did not dismiss that fact. What we state is that if the number of publications involving chromatography suffered such a notable increase because this is an important tool in science, otherwise its use, despite the generic growth in the number of publications, soon or later would be reflected in a decrease. However, considering the review comment and to avoid a miss interpretation of our view, we revised the text to convey the evolution of the different scientific fields and not only chromatography. Regarding the number of published articles, we think that the impact of the publications using separation techniques cannot be measured by the impact of Separation Science journals only because Separation Science alone have a limited impact. Instead, chromatographic techniques are being combined with other analytical techniques and cutting-edge applications that go beyond the Separation Science journals. This can be easily observed in the majority of the works published in Q1 journals (high impact factor) in the food, environmental, forensic, and analytical, among others, using high-resolution chromatography techniques (GC-MS; APGC-MS, LC-MS, UPLD-MS, UPLC-QToFMS, …). In other words, the high impact factors of journals not exclusively associated with Separation Science are also due to the excellent performance of the techniques used in Separation Science.

Reviewer#1 Along this line of argumentation, it is not clear according to which criterion the authors have selected the 118 references that are reported in this review (from the 20,000 to 25,000 chromatography-related publications that have appeared every year during at least the last decade).

AUTHORS ANSWER: A review article does not necessarily have to reach over 300 references. There are even journals that greatly limit the number of references allowed, including in review articles. The 118 references included in this review were selected because they are considered the most important or exemplificative to support the content of the article as well as the reports that are here discussed. Other reports and references, equally valid, could have been cited, but we think that having several references for works using the same analytical technique would not improve the review.

Reviewer#1 In any case, quoting (at least) 18 papers form the authors’ own production seems to put a bit too much emphasis on the own work.

Authors Answer: We agree that the inclusion of 18 papers out of 118 total references seems a little exaggerated. Therefore, we have replaced some of our references with others with similar works and research teams.

Reviewer#1 I have made a number of further – both lingual and scientific – comments directly into the pdf file of the manuscript.

AUTHORS ANSWER: We acknowledge the reviewer for helping us to significantly improve our manuscript.

Reviewer 2 Report

This manuscript focuses primarily on chromatography fundamentals, including the chromatography development process, chromatography-related concepts and classifications, challenges, and applications. I believe this manuscript will give readers engaged in analytical chemistry, especially beginners, a solid understanding of chromatography. In order to increase the readership of this paper, as the author has written extensively about the use of multidimensional chromatography and Nano-LC columns to improve chromatographic resolution and peak capacity, it is recommended that the manuscript will cover more high-interest sections on chromatographic separations, such as the chromatographic separation of analytes with high-polarity. There is an example in LC-MS/MS detection of organic acids in TCA cycle. Although many literatures have reported the separation of these organic acids using amino columns in HILIC mode, these separations were found to be less stable in practice. Is it possible to use capillary electrophoresis (CE) or ultra-high pressure ion chromatography (HPIC) to solve this problem? Also, in multi-dimensional LC, the robustness of the method is not very good. For example, the eluting portion of the first column was not fully compatible with the second column, resulting in peak dispersion and retention time shifts. Shimadzu reportedly has a technique to solve this problem (DOI: 10.1021/acs.analchem.1c03905).

Author Response

REVIEWER #2

Reviewer#2 This manuscript focuses primarily on chromatography fundamentals, including the chromatography development process, chromatography-related concepts and classifications, challenges, and applications. I believe this manuscript will give readers engaged in analytical chemistry, especially beginners, a solid understanding of chromatography. To increase the readership of this paper, as the author has written extensively about the use of multidimensional chromatography and Nano-LC columns to improve chromatographic resolution and peak capacity, it is recommended that the manuscript will cover more high-interest sections on chromatographic separations, such as the chromatographic separation of analytes with high-polarity. There is an example in LC-MS/MS detection of organic acids in the TCA cycle. Although many literatures have reported the separation of these organic acids using amino columns in HILIC mode, these separations were found to be less stable in practice. Is it possible to use capillary electrophoresis (CE) or ultra-high pressure ion chromatography (HPIC) to solve this problem? Also, in multi-dimensional LC, the robustness of the method is not very good. For example, the eluting portion of the first column was not fully compatible with the second column, resulting in peak dispersion and retention time shifts. Shimadzu reportedly has a technique to solve this problem (DOI: 10.1021/acs.analchem.1c03905).

AUTHORS ANSWER: We thank the reviewer comments for the improvement of the manuscript.  Multidimensional chromatography and nano-LC were presented in the MS, and explained their pertinence to the chromatography field within the goal of this MS. Indeed, multidimensional chromatography and nano-LC are presented as major achievements of the chromatography, and several examples of their application were included in the MS.

The example that the reviewer mention on the use of LC-MS/MS detection for the detection of the organic acids in the TCA cycle, was selected as a case study in the metabolomics area. Indeed, this case study was selected due to its interesting combination of LC-MS data with NMR and GC-MS data, using several body fluids. As it is stated on the MS, “instrumental conditions were adjusted according to the sample’ type (e.g. different chromatographic columns – SeQuant ZIC-cHILIC, BEH-C18 and HSS T3 (with <3µm particle size), different mobile phases, etc.)”. Nevertheless, the specific conditions were not fully described to make the MS less tedious and easy reading.

Also, in multi-dimensional LC, the robustness of the method is not very good. For example, the eluting portion of the first column was not fully compatible with the second column, resulting in peak dispersion and retention time shifts. Shimadzu reportedly has a technique to solve this problem.

AUTHORS ANSWER: A statement was added to the manuscript

Reviewer 3 Report

In this work, Câmara and co-workers give an overview of the application of chromatographic techniques in different areas, such as disease diagnosis, food safety or environmental pollution monitoring, among others. The topic of this review article is of interest due to the fundamental role that chromatographic techniques currently play in many areas. However, while this work has potential, some aspects should be taken into account before considering this work for publication in this journal.

- First of all, the introduction should be modified considerably. In this sense, section 1.2 deals with quite basic aspects of chromatographic techniques that are widely known, so it would be advisable to summarize them considerably or even eliminate them. It is more interesting to describe about the advances of these techniques that have made them position themselves and acquire the relevance they have today. In this sense, and although the main topic of the review focuses on chromatographic techniques, it is also essential to highlight the importance of advances in detection systems and how these have affected the development of chromatography. In addition, it is also important to highlight the fundamental role that sample preparation plays in guaranteeing good performance of chromatographic techniques in the analysis of complex samples or samples that are incompatible with the system.

- In general, when it is said that the introduction of modifications in the techniques, whether in the dimensions of the column, the characteristics of the stationary phase or in any other aspect, has allowed their range of application to be extended, it would be interesting to describe these modifications in a general way in order to provide clearer information in that regard (e.g., lines 510-513).

- In general, and honestly speaking, there are parts of the manuscript that are very dense and should be summarized or modified. This is especially noticeable in section 3 of the manuscript, which is probably the most important part of the work. An example of this would be the description of some previously published articles (for example, between lines 547 and 571). Instead, section 3 should provide more general information on which techniques are the most widely used in each field and the most relevant conditions, simply commenting on the most significant aspects of some works, difficulties related to the analysis of this type of matrices, etc. The paragraph between lines 654 and 678 could be taken as a reference.

- To all of the above should be added the inclusion of a table that collects the most important aspects of the works mentioned in the text. This would allow the reader to have a global and direct vision of the different applications of chromatographic techniques in the different areas considered in this review.

Other minor comments:

- I suggest a revision of the article title, as it does not fully reflect the content of the manuscript.

- I suggest that the manuscript be reviewed by a native English speaker, as it contains some grammatical errors that should be corrected.

- I would suggest a revision of the keywords to avoid repetition and provide a more general view of the main themes developed in the manuscript. For example, "types of chromatography" could be removed.

- Line 101. Developed.

- Line 109. “… (TLC), and…”.

- I would suggest authors avoid the use of “!” in the manuscript.

- Considering that HPLC is used throughout the manuscript with the current meaning of "high performance liquid chromatography", it should be written when the term is mentioned for the first time (lines 124-125).

- In general, the abbreviations should be reviewed, as some of them have not been defined while others have been defined more than once.

- The space between numbers and their units is often missing. It should be corrected.

- Line 257. The mobile phase can also include water, for example.

- Line 293. HPLC separations.

- If "ultra-high-performance liquid chromatography" has been abbreviated as UHPLC, use this abbreviation instead of UPLC.

- In general, I would suggest standardizing the nomenclature of the instrumentation used, especially when MS detectors are mentioned. If the ionization source and analyser are mentioned, they should always be mentioned.

- Line 843. Sensitivity.

- Figures with higher resolution should be provided.

Author Response

REVIEWER #3

Reviewer#3 In this work, Câmara and co-workers give an overview of the application of chromatographic techniques in different areas, such as disease diagnosis, food safety and environmental pollution monitoring, among others. The topic of this review article is of interest due to the fundamental role that chromatographic techniques currently play in many areas. However, while this work has potential, some aspects should be taken into account before considering this work for publication in this journal.

- First of all, the introduction should be modified considerably. In this sense, section 1.2 deals with quite basic aspects of chromatographic techniques that are widely known, so it would be advisable to summarize them considerably or even eliminate them. It is more interesting to describe the advances of these techniques that have made them position themselves and acquire the relevance they have today. In this sense, and although the main topic of the review focuses on chromatographic techniques, it is also essential to highlight the importance of advances in detection systems and how these have affected the development of chromatography. In addition, it is also important to highlight the fundamental role that sample preparation plays in guaranteeing the good performance of chromatographic techniques in the analysis of complex samples or samples that are incompatible with the system.

AUTHORS ANSWER: According to the reviewer comment, more information regarding detection systems, as well as sample preparation was added to the manuscript (Lines 232-240; 266-276; 329-336).

Reviewer#3 In general, when it is said that the introduction of modifications in the techniques, whether in the dimensions of the column, the characteristics of the stationary phase or any other aspect, has allowed their range of application to be extended, it would be interesting to describe these modifications in a general way to provide clearer information in that regard (e.g., lines 510-513).

AUTHORS ANSWER: According to the reviewer suggestion the statement was rearranged as follows: “ … This is exemplified by the huge power of two-dimensional gas chromatography (2D-GC commonly known as GC × GC)) and two-dimensional liquid chromatography (2D-LC), for separating complex mixtures. The main concept in 2D chromatography is the use of two separate columns with two different stationary phases. The effluent from the 1D column is injected into a 2D column where occurs the separation of all the unresolved analytes present in the 1D effluent is based on the differences in selectivity of the two dimensions. “

Reviewer#3 In general, and honestly speaking, there are parts of the manuscript that are very dense and should be summarized or modified. This is especially noticeable in section 3 of the manuscript, which is probably the most important part of the work. An example of this would be the description of some previously published articles (for example, between lines 547 and 571). Instead, section 3 should provide more general information on which techniques are the most widely used in each field and the most relevant conditions, simply commenting on the most significant aspects of some works, difficulties related to the analysis of this type of matrices, etc. The paragraph between lines 654 and 678 could be taken as a reference.

AUTHORS ANSWER: The purpose of section 3 is to present specific case studies that highlight the importance of selecting appropriate chromatographic methods and equipment for the analyses of different types of analytes, as stated in MS. In this way, readers will be able to understand how the data obtained from the chromatographic analysis can be essential to respond to current societal challenges, with a huge impact on our health and well-being, as well as on the valorisation of natural resources and sustainability. Therefore, three main areas of investigation were selected, within each, several case studies were explored. Each case study describes its goals, information regarding the chromatographic analysis (including instrumental details), and consequently the main achieved results. In this sense, reader may encounter different approaches, and for instance how can combine different chromatographic techniques. Thus, the general information is previously provided to the readers in other sections, while section 3 provides practical examples of those techniques.

Reviewer#3 Other minor comments:

- I suggest a revision of the article title, as it does not fully reflect the content of the manuscript.

- I suggest that the manuscript be reviewed by a native English speaker, as it contains some grammatical errors that should be corrected.

- I would suggest a revision of the keywords to avoid repetition and provide a more general view of the main themes developed in the manuscript. For example, "types of chromatography" could be removed.

AUTHORS ANSWER: According to the reviewer suggestions, the corrections/alterations were done in the revised version of the MS.

Reviewer#3

- Line 101. Developed.

- Line 109. “… (TLC), and…”.

- I would suggest authors avoid the use of “!” in the manuscript.

AUTHORS ANSWER: According to the reviewer suggestions, these corrections were done in the revised version of the manuscript.

Reviewer#3 Considering that HPLC is used throughout the manuscript with the current meaning of "high-performance liquid chromatography", it should be written when the term is mentioned for the first time (lines 124-125).

AUTHORS ANSWER: Corrected, according to reviewer suggestion.

Reviewer#3 In general, the abbreviations should be reviewed, as some of them have not been defined while others have been defined more than once.

- The space between numbers and their units is often missing. It should be corrected.

- Line 257. The mobile phase can also include water, for example.

- Line 293. HPLC separations.

- If "ultra-high-performance liquid chromatography" has been abbreviated as UHPLC, use this abbreviation instead of UPLC.

AUTHORS ANSWER: UPLC was used in line 579 to describe the instrument used, not the technique. So in this case, the UPLC designation was correctly used. Otherwise, we agree that UHPLC should be employed to designate the ultra-high-performance liquid chromatography technique and we have revised the MS accordingly.

Reviewer#3 In general, I would suggest standardizing the nomenclature of the instrumentation used, especially when MS detectors are mentioned. If the ionization source and analyser are mentioned, they should always be mentioned.

AUTHORS ANSWER: The abbreviations were revised throughout the MS and a list of abbreviations was added.

Reviewer#3 Line 843. Sensitivity.

AUTHORS ANSWER: corrected.

Reviewer#3 Figures with higher resolution should be provided.

AUTHORS ANSWER: added.

AUTHORS ANSWER: According to the reviewer suggestions, all the above corrections/alterations were done in the revised version of the manuscript.

Reviewer 4 Report

The review titled "Chromatographic-based platforms: new avenues for science 2 progress and sustainability" covers an interesting topic. However, In my opinion, I believe that the content and structure of the document is not suitable for a review in Molecules journal. A review should focus on the current state of knowledge of a topic. The topic covered in this review is very broad. In addition, the content is also treated in a very didactic way (history of the LC, concepts, ...). Although the content is totally correct, I consider that this type of publication should be more oriented to a textbook or book chapter than to a review. In addition, the current applications today of chromatographic techniques (for example: omics, ...) are thanks to the use coupled to MS. Therefore, a review in this topic should focus cover the recent advances of LC-MS, GC-MS or CE-MS in a certain application. Trying to cover all the chromatographic techniques in a multitude of applications does not fit with the philosophy of a review. 

Author Response

REVIEWER #4

Reviewer#4 The review titled "Chromatographic-based platforms: new avenues for science 2 progress and sustainability" covers an interesting topic. However, In my opinion, I believe that the content and structure of the document are not suitable for a review in Molecules journal. A review should focus on the current state of knowledge of a topic. The topic covered in this review is very broad. In addition, the content is also treated in a very didactic way (history of the LC, concepts, ...). Although the content is correct, I consider that this type of publication should be more oriented to a textbook or book chapter than to a review. In addition, the current applications today of chromatographic techniques (for example omics, ...) are thanks to the use coupled to MS. Therefore, a review on this topic should focus cover the recent advances of LC-MS, GC-MS or CE-MS in a certain application. Trying to cover all the chromatographic techniques in a multitude of applications does not fit with the philosophy of a review.

AUTHORS ANSWER: We acknowledge and respect the reviewer's comment and opinion. However, we think that despite most reviews trying to cover the most recent developments or knowledge in a given topic, this should not be a condition to not consider another type of review article. For example, a review article based on theoretical or historical fundamentals cannot focus on the current state of knowledge and is still a valid review article. Similarly, we submitted a review that covers the fundamentals and evolution of chromatography and ends with the discussion of several recent works using cutting-edge chromatographic techniques, which incorporate the current state of knowledge of chromatography. Many other works could have been mentioned, but no matter how many more we included, there would be many more to be mentioned. We have mentioned those reports that we consider that made a difference in terms of the importance of chromatography for the evolution of science and to help solve the most important societal challenges.

Round 2

Reviewer 1 Report

The authors are thanked for carefully and to a large extent revising their manuscript according to the comments of the referee(s).

They are also thanked for the detailed comments on the report of referee #1. While not in all aspects completely convincing, they manage still to make their point.

As the technical side of the mansucript has improved largely, this mansucript is considered acceptable for publication in the present form.

Reviewer 3 Report

After reviewing the second version of the manuscript by Câmara et al., it can be seen that the authors have hardly taken into account the comments of the reviewers, making minor changes in a work that required, in many sections, a complete restructuring. In fact, some comments have not even been considered or responded to by the authors, such as the inclusion of a table that collected the most important aspects of all the works discussed in the manuscript. On the other hand, some of the parts that have been added are not properly linked to the above, nor are references even provided to support this new content, the level of some sections is still too basic for a review of this type and relevant information is still missing.

Taking into account the above, this article cannot be considered for publication in this journal.